# GABA_A_ and Glycine Receptor-Mediated Inhibitory Synaptic Transmission onto Adult Rat Lamina II_i_ PKCγ-Interneurons: Pharmacological but Not Anatomical Specialization

**DOI:** 10.3390/cells11081356

**Published:** 2022-04-15

**Authors:** Corinne El Khoueiry, Cristina Alba-Delgado, Myriam Antri, Maria Gutierrez-Mecinas, Andrew J. Todd, Alain Artola, Radhouane Dallel

**Affiliations:** 1Neuro-Dol, Inserm, Université Clermont Auvergne, CHU Clermont-Ferrand, F-63000 Clermont-Ferrand, France; corinekhoueiry90@hotmail.com (C.E.K.); cristina.alba_delgado@uca.fr (C.A.-D.); myriam.antri@uca.fr (M.A.); 2Institute of Neuroscience and Psychology, University of Glasgow, Glasgow G12 8QQ, UK; maria.gutierrez-mecinas@glasgow.ac.uk (M.G.-M.); andrew.todd@glasgow.ac.uk (A.J.T.)

**Keywords:** mechanical allodynia, medullary dorsal horn, protein kinase C gamma, inhibitory interneurons, spontaneous IPSCs, miniature IPSCs, α2 subunit of GABA_A_ receptor, α1 subunit of glycine receptor, gephyrin, glutamic acid decarboxylase, glycine transporter 2

## Abstract

Mechanical allodynia (pain to normally innocuous tactile stimuli) is a widespread symptom of inflammatory and neuropathic pain. Spinal or medullary dorsal horn (SDH or MDH) circuits mediating tactile sensation and pain need to interact in order to evoke mechanical allodynia. PKCγ-expressing (PKCγ^+^) interneurons and inhibitory controls within SDH/MDH inner lamina II (II_i_) are pivotal in connecting touch and pain circuits. However, the relative contribution of GABA and glycine to PKCγ^+^ interneuron inhibition remains unknown. We characterized inhibitory inputs onto PKCγ^+^ interneurons by combining electrophysiology to record spontaneous and miniature IPSCs (sIPSCs, mIPSCs) and immunohistochemical detection of GABA_A_Rα2 and GlyRα1 subunits in adult rat MDH. While GlyR-only- and GABA_A_R-only-mediated mIPSCs/sIPSCs are predominantly recorded from PKCγ^+^ interneurons, immunohistochemistry reveals that ~80% of their inhibitory synapses possess both GABA_A_Rα2 and GlyRα1. Moreover, nearly all inhibitory boutons at gephyrin-expressing synapses on these cells contain glutamate decarboxylase and are therefore GABAergic, with around half possessing the neuronal glycine transporter (GlyT2) and therefore being glycinergic. Thus, while GABA and glycine are presumably co-released and GABA_A_Rs and GlyRs are present at most inhibitory synapses on PKCγ^+^ interneurons, these interneurons exhibit almost exclusively GABA_A_R-only and GlyR-only quantal postsynaptic inhibitory currents, suggesting a pharmacological specialization of their inhibitory synapses.

## 1. Introduction

Mechanical allodynia (pain sensation in response to a normally innocuous mechanical stimulus) is one of the most prevalent symptoms of both inflammatory (initiated by tissue damage/inflammation) and neuropathic (following nervous system lesion) chronic pain syndromes [1,2]. How can light touch evoke pain? It is now well established that mechanical allodynia is associated with the activation, within the spinal dorsal horn (SDH) or medullary dorsal horn (MDH), of dorsally directed polysynaptic circuits. These allow tactile inputs (Aβ-, Aδ-, and C-fiber low-threshold mechanoreceptors, LTMRs), which terminate within or below inner lamina II (II_i_) [3,4], to gain access to the pain circuitry in the superficial SDH/MDH [5,6,7,8]. Mechanical allodynia thus results from a miscoding, with cells that normally respond to and transmit only noxious stimuli being activated by tactile inputs.

Key elements for such circuits are interneurons within lamina II_i_ that express the γ isoform of protein kinase C (PKCγ) [5,6,9,10,11]. Direct evidence for the activation of lamina II_i_ PKCγ^+^ interneurons under mechanical allodynia comes from studies of PKCγ immunoreactivity [12,13] and anatomical markers of neuronal activation (Fos protein and pERK1/2) [5,6,14,15,16,17]. Moreover, genetic [10,18] or pharmacological inactivation of PKCγ [5,6,14,16] prevents mechanical allodynia, whereas its activation is sufficient to induce it [16]. Together with the evidence that lamina II_i_ PKCγ^+^ interneurons directly receive tactile inputs [19,20,21,22,23,24], this strongly suggests that activation of lamina II_i_ PKCγ^+^ interneurons is pivotal in conveying LTMR inputs to the superficial SDH/MDH pain circuitry.

It is assumed that LTMR inputs onto these PKCγ^+^ interneurons activate convergent feedforward inhibitory circuits that prevent such LTMR inputs from activating PKCγ^+^ interneurons and, in turn, from being transferred to pain circuitry. Allodynia is thought to stem from the loss of feedforward inhibition onto these PKCγ^+^ interneurons [5,6,14]. This conclusion is consistent with recent circuit-based evidence. Using paired recordings, Lu et al. [20] showed that PKCγ^+^ interneurons are normally under feedforward glycinergic disynaptic inhibition from lamina III neurons, thus preventing Aβ LTMRs from activating PKCγ^+^ interneurons. PKCγ^+^ interneurons are also inhibited by parvalbumin-expressing inhibitory interneurons [15], which receive strong monosynaptic input from several classes of myelinated LTMRs [25]. Notably, these inhibitory connections onto lamina II_i_ PKCγ^+^ interneurons appear to be lost under neuropathic conditions [15,20].

Both glycine (GlyR) and GABA_A_ receptors (GABA_A_R) are present on lamina II_i_ PKCγ^+^ interneurons [5,22]. However, functional evidence suggests a pharmacological specialization of inhibitory inputs onto the lamina II_i_ PKCγ^+^ interneurons. Thus, Aβ LTMR inputs onto these PKCγ^+^ interneurons were shown to be under exclusively feedforward glycinergic inhibition [20,26]. Moreover, mechanical allodynia is specifically dynamic under glycinergic disinhibition but static under GABA_A_ergic disinhibition [5,6,14]. This issue has already been addressed for lamina II neurons with whole-cell patch-clamp recordings in ex vivo SDH/MDH slices and analysis of action potential-independent miniature inhibitory postsynaptic currents (mIPSCs). Such studies revealed that, whereas GABA and glycine are co-released from interneurons [27,28,29] and GABA_A_R and GlyR are colocalized on postsynaptic neurons [30], quantal postsynaptic currents are either GlyR- or GABA_A_R-mediated, but never both, in adult neurons [27,28,31,32,33,34,35,36,37]. Importantly, in all these studies, the identities of the recorded neuron were never examined. Therefore, whether such a pharmacological specialization of inhibitory synapses exists in the lamina II_i_ PKCγ^+^ interneurons and whether it is related to any anatomical specialization are still unknown.

To address these issues, we carried out (i) an electrophysiological analysis (whole-cell patch-clamp recordings) of spontaneous IPSCs (sIPSCs) and mIPSCs recorded from PKCγ^+^ interneurons in the adult rat MDH and (ii) an immunocytochemical analysis of GlyRs and GABA_A_Rs, present at inhibitory synapses on these cells. We found that three distinct populations of quantal inhibitory synaptic currents can be recorded from MDH lamina II_i_ PKCγ^+^ interneurons according to decay kinetics: fast monoexponential, slow monoexponential, and biexponential mIPSCs. Notably, all PKCγ^+^ interneurons exhibit these three types of mIPSCs in equal proportions. Pharmacological analysis indicates that fast monoexponential mIPSCs are GlyR-mediated, and slow monoexponential ones are GABA_A_R-mediated. Therefore, most, if not all, mIPSCs onto MDH lamina II_i_ PKCγ^+^ interneurons are either GlyR- or GABA_A_R-mediated, not both. On the other hand, our immunocytochemical study provides evidence in favor of colocalization of GABA_A_R and GlyR on MDH lamina II_i_ PKCγ^+^ interneurons. Altogether, this allows for a functional specialization of the feedforward inhibition of the different LTMR inputs onto PKCγ^+^ interneurons (see the Figure 1 in the work of [9]) but raises the question as to how anatomical data can account for such specialization.

## 2. Materials and Methods

### 2.1. Animals

Adult male Sprague–Dawley rats (21–35 days old, 50–100 g, *n* = 13; 42–49 days old, 280–290 g, *n* = 3) were obtained from Charles River (L’Arbresle, France) and Harlan (Loughborough, UK) Laboratories, respectively, and housed 3–4 per cage under standard laboratory conditions (22 ± 1 °C, 12 h light/dark cycle, lights on at 07:00 a.m., food and water ad libitum). All efforts were made to minimize the number of animals used. All animal experiments were performed in accordance with the ethical guidelines of the International Association for the Study of Pain (IASP) [38], the Directive 2010/63/UE of the European Parliament, the Council on the Protection of Animals Used for Scientific Purpose, and the UK. Animals (Scientific Procedures) Act 1986. Protocols applied in this study were approved by the local animal experimentation committees: CEMEAA “Comité d’Ethique en Matière d’Expérimentation Animale Auvergne” (n° CE 28-12) and the Ethical Review Process Applications Panel of the University of Glasgow.

### 2.2. Electrophysiology Experiments

#### 2.2.1. Slice Preparation

All procedures were conducted between 9:00 a.m. and 4:00 p.m. Rats from 21 to 35 days (*n* = 13) were deeply anesthetized with an intraperitoneal (i.p.) overdose of chloral hydrate (7%) and then quickly decapitated. The whole brain, including the upper cervical region of the spinal cord, was carefully removed and placed into cold (4 °C) sucrose-based artificial cerebrospinal fluid (aCSF) containing (in mM): 205 sucrose, 2 KCl, 7 MgCl_2_, 26 NaHCO_3_, 1.2 NaH_2_PO_4_, 11 d-glucose, and 0.5 CaCl_2_ (pH 7.4) bubbled with 95% O_2_ and 5% CO_2_. Serial transverse and parasagittal slices (350 µm thick) were cut from the brainstem using a vibrating-blade microtome (VT1200 S, Leica Microsystèmes SAS, Nanterre, France). Slices were incubated at 37 °C in aCSF containing (in mM): 130 NaCl, 3 KCl, 2.5 CaCl_2_, 1.3 MgSO_4_, 0.6 NaH_2_PO_4_, 25 NaHCO_3_, 10 glucose (pH 7.4) bubbled with 95% O_2_ and 5% CO_2_, for a 60 min recovery period.

#### 2.2.2. Patch-Clamp Recordings

Slices were transferred into a recording chamber (volume ≈ 1 mL) and held down with a C-shape platinum wire supporting transverse nylon fibers. The chamber is mounted on an upright microscope fitted with fluorescence optics (AxioExaminer, Carl Zeiss, Hamburg, Germany) and linked to a digital camera QImaging Exi Aqua (Ostrava, Czech Republic). Slices were continuously perfused at 3.0 mL/min with aCSF solution maintained at room temperature (≈25 °C). The substantia gelatinosa of MDH has a distinct translucent appearance that can be easily distinguished under the microscope using the 10× objective lens. Lamina II can be divided into outer and inner (II_i_) laminae II of equal size, and neurons were recorded within lamina II_i_ under visual control using a 63× water-immersion objective lens with combined infrared and differential interference contrast (DIC).

Patch pipettes were pulled from borosilicate glass tubing (Sutter Instrument, Novato, CA, USA). To study IPSCs, the pipettes were filled with an intracellular solution containing (in mM): 145 KCl, 5 EGTA, 2 MgCl_2_, 10 Hepes, 2 ATP-Na_2_, 0.2 GTP-Na_2_ neurobiotin (0.05%, Vector Laboratories, Burlingame, CA, USA), dextran tetramethylrhodamine (10,000 MW, fluoro-ruby, 0.01%, Life technologies, Saint Aubin, France), pH adjusted to 7.4, and osmolarity of 290–300 mOsm. Pipette resistances ranged from 5 to 7 MΩ. Recorded neurons were maintained at a holding potential of −65 mV. Under the above conditions, the reversal potential for Cl^−^ ions is close to 0 mV, and IPSCs are thus inward currents at the holding potential.

Spontaneous and miniature IPSCs (sIPSCs and mIPSCs, respectively) were recorded using the patch-clamp technique in whole-cell configuration and voltage-clamp mode. Acquisitions were performed using Clampex 10 software (Molecular Devices, Sunnyvale, CA, USA) connected to a Multiclamp 700B amplifier (Molecular Devices, Sunnyvale, CA, USA) via a Digidata 1440A digitizer (Molecular Devices, Sunnyvale, CA, USA), or Patch Master v2x65 software (HEKA Elektronik, Lambrecht, Germany) connected to an EPC10 amplifier (HEKA Elektronik, Lambrecht, Germany). Voltage-clamp data were low pass filtered at 3 kHz and digitized at 10 kHz. Series resistance was monitored throughout the experiments and was not compensated. Data were discarded if the series resistance varied more than ±20 MΩ.

#### 2.2.3. Drug Application

To block glutamatergic neurotransmission, slices were continually perfused with oxygenated ACSF containing 6-cyano-7-nitroquinoxaline-2,3-dione (CNQX; 10 µM; Tocris Cookson, Ballwin, MO, USA) and D2-amino-5-phosphonovaleric acid (APV; 40 µM; Tocris Cookson). For mIPSC recordings, tetrodotoxin (TTX; 0.5 µM; Sigma) was added to the bath. Selective antagonists were used to block glycine receptors (strychnine hydrochloride; 0.5 µM) and GABA_A_ receptors (bicuculline methiodide; 10 µM; Research Biochemicals).

#### 2.2.4. Immunofluorescence Detection of PKCγ/Neurobiotin Interneurons

At the end of the recordings, epifluorescence was used to ensure that the recorded cells were filled with neurobiotin. Slices were then transferred into 4% paraformaldehyde in 0.1 M phosphate-buffered solution (pH 7.4) and stored overnight at 4 °C. Double labeling PKCγ/neurobiotin was performed for *post-hoc* analysis. Slices were first incubated with Avidin DCS-rhodamine (1:200, Vector Laboratories, Burlingame, CA, USA) and then placed into primary antibody solution containing polyclonal guinea pig anti-PKCγ (1:4000; Frontier Institute Co., Ltd, Hokkaido, Japan; RRID: AB-2571826). After subsequent washes, slices were incubated with Alexa Fluor^®^ 488-conjugated goat anti-guinea pig secondary antibody (1:200, Jackson Immunoresearch Laboratories, Inc., West Grove, PA, USA; RRID: AB-2337438). Slices were then transferred onto gelatinized slides and dehydrated before being coverslipped with DPX mountant for histology. Double-immunolabeling was examined with a Zeiss LSM 510 confocal laser scanning microscope (Carl Zeiss, Hamburg, Germany) by using 488 and 532 nm excitation laser light in the original thick slices. In order to suppress emission cross-talk, the microscope was configured to perform all scanning in sequential mode. Z-series (z-step of 0.38 µm) were scanned at x40 magnification with an oil-immersion lens. Neurite reconstruction and morphological classification of neurons were performed as previously described in the work of [39].

#### 2.2.5. Data Analysis

IPSCs were analyzed offline using Clampfit 10.0 software (Axon Instrument, Molecular Devices, Sunnyvale, CA, USA) and Electrophysiology Data Recorder/Whole-Cell Analysis Program (WinEDR/WinWCP; Dr. J. Dempster). Events were detected and analyzed over a 10 min-long data segment using an automated low-amplitude (−10 pA, duration 3 ms) threshold detection algorithm. Each event was visually examined, and any noise that spuriously met trigger specifications was rejected. Several characteristics of the accepted events were analyzed: peak amplitude, instantaneous frequency, inter-event interval, and rise and decay tau (*τ*). A least-squares minimization algorithm was used to determine the decay *τ* of IPSCs. The decay phase of individual IPSCs was fitted (90%–10% of the peak amplitude) by either a monoexponential (*y*(*t*) = *A*e−*t*/*τ*) or biexponential (*y*(*t*) = *A*fast (−*t*/*τ* fast) + *A*slow(−*t*/*τ* slow)) function, where *t* is time, *A* amplitude, and *τ* the decay *τ*. An improvement of the fit by two exponentials compared to one resulted in a significant reduction in the standard deviation of the residuals, as confirmed by the use of the *F*-test. Traces containing multiple events were discarded, and only events that had stable baselines before the rise and after the end of the decay were kept for analysis. For IPSC averaging, the initial rising phases of the successive events were artificially aligned using software.

### 2.3. Histological Procedures

#### 2.3.1. Immunofluorescence Detection of Inhibitory Receptors

For detecting the expression of GABA_A_Rs and GlyRs on PKCγ interneurons, rats (280–290 g) were deeply anesthetized (pentobarbitone: 300 mg i.p.) and transcardially perfused with freshly prepared 4% formaldehyde. Transverse sections (60 μm thick) through the MDH were obtained with a vibrating-blade microtome (VT1200S, Leica). To perform the triple immunostaining, the immunoreaction with anti-PKCγ was first revealed with tyramide signal amplification (TSA) to stabilize the fluorescent staining of PKCγ interneurons and prevent it from being washed out during subsequent processing (with pepsin). Sections were initially incubated in rabbit anti-PKCγ (Santa Cruz Biotechnology, Santa Cruz, CA, USA, catalog number sc-211; 1:20,000; RRID: AB-632234) for 3 days and then overnight in donkey anti-rabbit IgG conjugated to horseradish peroxidase (Jackson Immunoresearch; RRID: AB-10015282). Immunoreactivity was revealed with a TSA kit (tetramethylrhodamine; PerkinElmer Life Sciences, Boston, MA, USA). Tissues were then incubated for 30 min at 37 °C in 0.2 M HCl containing 0.25 mg/mL pepsin to unmask antigenic sites [40]. Finally, a second immunoreaction was performed by incubating sections for 2 days with rabbit anti-GlyRα1 (anti-GlyRα1 catalog number 146,111, Synaptic Systems; 1:1000; RRID: AB-887723) and guinea pig anti-GABA_A_α2 subunit (a gift from J-M Fritschy; 1:1000; RRID: AB-2314463), which were revealed with species-specific secondary antibodies raised in donkey and conjugated to DyLight 649 (Jackson Immunoresearch; RRID: AB-2315775) and Alexa488 (Jackson Immunoresearch; RRID: AB-2340472), respectively. Sections were mounted in anti-fade medium and stored at −20 °C.

We chose the guinea pig anti-GABAAα2 receptor antibody for two reasons: firstly, because it tolerated the pepsin treatment, which was required for antigen retrieval, and secondly, because it was raised in a different species from the anti-GlyRα1 antibody. Although early in situ hybridization studies failed to detect mRNA for the GABA_A_α2 subunit in the spinal dorsal horn [41], in recent transcriptomic studies [42,43], the mRNA (Gabra2) has been found in many neurons in the dorsal horn of the mouse, including excitatory interneurons that express PKCγ. In addition, in situ hybridization studies using the very sensitive RNAscope method have revealed that many excitatory neurons in the superficial dorsal horn of the rat spinal cord contain Gabra2 mRNA [44,45].

#### 2.3.2. GABAergic and Glycinergic Boutons on PKCγ Interneurons

Transverse sections (70 μm thick) through the MDH from the same 3 male Sprague–Dawley rats were immersed in 50% ethanol for 30 min to facilitate antibody penetration [46]. Immunofluorescent staining for the glycine transporter GlyT2, glutamic acid decarboxylase GAD (both isoforms), gephyrin, and PKCγ was carried out by incubating sections for 2 days at 4 °C in the following mixture of primary antibodies: goat anti-PKCγ (a gift from M Watanabe; 1:1000), rabbit anti-GlyT2 (a gift from F Zafra; 1:1000), mouse monoclonal anti-GAD65 (GAD6, Developmental Studies Hybridoma Bank, University of Iowa; 1:100), and mouse monoclonal anti-GAD67 (Merck, Watford, UK, catalog number mAb5406; 1:5000). These were revealed by overnight incubation in species-specific secondary antibodies raised in donkey and conjugated to Rhodamine Red, Alexa488, and Alexa647 (Jackson Immunoresearch), respectively. To avoid binding of the mouse secondary antibody to primary antibodies used in the subsequent step, a Fab’ fragment of donkey anti-mouse IgG (conjugated to Alexa 647) was used. The sections were then incubated overnight in mouse monoclonal antibody against gephyrin (clone 7a), conjugated to the fluorescent dye Oyster550 (catalog number 147 011C3, Synaptic Systems; 1:500), rinsed, and mounted in anti-fade medium and stored at −20 °C.

#### 2.3.3. Confocal Microscopy and Analysis

Sections were scanned with a Zeiss LSM710 confocal microscope with argon multi-line, 405 nm diode, 561 nm solid-state, and 633 nm HeNe lasers. For analysis of GABA_A_ and glycine receptor expression, confocal image stacks consisting of 15–27 optical sections at 0.5 μm z-separation were acquired from the MDH of the 3 rats (6–9 scans per animal) by scanning through a 63× oil-immersion lens (numerical aperture 1.4). Sections were analyzed with Neurolucida for Confocal software (MBF Bioscience, Williston, VT, USA). Twenty-five PKCγ-immunoreactive (IR) cells were identified and selected in scans from each animal before the distribution of receptor staining was viewed. Staining for the GABA_A_α2 subunit was then viewed, and the locations of all immunoreactive puncta on the cell bodies of the selected PKCγ cells were noted. Staining for GlyRα1 was then viewed, and the presence or absence of this subunit at each GABA_A_α2 punctum was recorded. In addition, a search was performed for GlyRα1-positive puncta that lacked GABA_A_α2-immunoreactivity. A separate analysis was performed to determine whether the relationship between the two receptor subunits was similar in the dendrites of PKCγ neurons. For this analysis, we initially viewed the PKCγ and GABA_A_α2 channels and selected ~100 GABA_A_α2-IR puncta that were located on PKCγ dendrites in sections from each of the 3 rats. We then viewed the GlyRα1 channel and noted the presence or absence of this type of immunoreactivity on each of the selected GABA_A_α2 puncta. We subsequently performed the reverse analysis by viewing PKCγ and GlyRα1 channels and selecting ~100 GlyRα1-immunoreactive puncta that were on PKCγ dendrites from each rat. The GABA_A_α2 channel was then viewed, and the presence or absence of GABA_A_α2 immunostaining was noted for each GlyRα1 punctum.

To analyze the relationship between GAD, GlyT2, gephyrin, and PKCγ, the MDH of the 3 rats was scanned (between 1 and 6 sections were scanned through the x63 lens, z-step 0.3 μm). Gephyrin-IR puncta on the cell bodies or dendrites of PKCγ cells were initially identified (73–86 puncta per animal, mean 77.7). The channels corresponding to GADs and GlyT2 were then viewed, and the presence or absence of staining for each of these was noted for all of the selected gephyrin puncta.

#### 2.3.4. Characterization of Antibodies

The guinea pig and goat antibodies against PKCγ were raised against amino acids 684–697 of the mouse protein and recognized a single protein band of 75 kDa on immunoblots (manufacturer’s specification). The rabbit anti-PKCγ has been shown to stain identical structures to the guinea pig antibody [47]. The GABA_A_α2 antibody, raised against amino acids 1–9, recognizes a single band of 52 kDa on the Western blot of mouse cerebral cortex [48]. The GlyRα1 antibody was raised against a recombinant protein corresponding to amino acids 1 to 457 from rat glycine receptor α1 and is specific for the N-terminal 10 residues of this subunit [49]. The gephyrin antibody detects the brain-specific 93 kDa splice variant, and specificity has been verified in knock-out tissue (manufacturer’s specification). The GlyT2 antibody was raised against amino acids 1–193 and detected a 90–110 kDa band in immunoblots [50]. Both monoclonal GAD antibodies are highly selective for the corresponding isoform [51] (manufacturer’s specification).

### 2.4. Statistical Analysis

All statistical analyses were performed with the software GraphPad Prism (v7.1, La Jolla, CA, USA). Continuously distributed data were displayed either by showing all data points or by using box-and-whisker plots with all elements (mean, median, interquartile interval, minimum, maximum). Categorical variables were described by relative frequencies. The minimal sample size required to detect a standardized effect size of at least 20% with a power analysis >70% was calculated and respected as much as possible. The normality distribution of data was determined using Kolmogorov–Smirnov tests. The comparison between groups was made using one-way or two-way analysis of variance (ANOVA) followed by Tukey’s *post-hoc* test for multiple comparisons. F-values were expressed with their associated degrees of freedom (df). Factors were designed as follows: *IPSC type* for comparisons between fast monoexponential, slow monoexponential, and biexponential mIPSC or sIPSC, *Cell type* for comparisons between PKCγ^+^ and PKCγ, and *Interaction* for effect between two factors on the dependent variable. Grubbs’ test was used to identify outlier data. The level of significance was set at *p* < 0.05. Statistical details for each quantitative experiment were illustrated in Table 1 and Table 2. Figures were created using GraphPad Prism, CorelDraw Graphics (v12.0, Ottawa, Canada), or Photoshop (vCS6 Adobe Systems Incorporated, San Jose, CA, USA) software.

## 3. Results

### 3.1. Electrophysiological Analysis

We recorded spontaneous IPSCs, in the presence (i.e., mIPSCs) or absence of TTX (i.e., sIPSCs), from lamina II_i_ neurons under both CNQX (10 µM) and APV (40 µM) in the bath. To be included in this analysis, each recorded neuron had to be successfully filled with neurobiotin, be PKCγ phenotyped (Figure 1(A1)), and show a sufficiently high frequency of sIPSCs or mIPSCs to permit analysis. A total of 10 PKCγ^+^ interneurons and 10 PKCγ^−^ interneurons fulfilled such criteria.

#### 3.1.1. Three Kinetically Distinct Populations of mIPSCs onto MDH Lamina II_i_ PKCγ^+^ Interneurons

It is now well established that GlyR- and GABA_A_R-mediated evoked IPSCs or mIPSCs recorded from lamina I–II neurons in the MDH [32] as well as SDH [27,32,33,35,37,52] can be distinguished on the basis of their decay kinetics, the decay *τ* of GlyR-mediated IPSCs being on average a factor of 2.4 [35], 3.6 [28], or 5.0 [27] faster than that of GABA_A_R-mediated IPSCs. Therefore, to assess the nature of inhibitory synaptic inputs onto MDH lamina II_i_ PKCγ^+^ interneurons, we first examined the decay kinetics of mIPSCs recorded from 5 PKCγ^+^ interneurons. Heterogeneous kinetics of mIPSCs were identified (Figure 1B). The decay time courses of most mIPSCs were best fitted by monoexponential functions. Close inspection of the distribution of the decay *τ* of these monoexponential mIPSCs revealed a heterogeneous population of decays, which were best fitted by the sum of two Gaussians (Figure 1C). One population of monoexponential mIPSCs exhibited a fast decay phase (peak of decay *τ*: 8.9 ± 0.9 ms; Figure 1D). The other population of monoexponential mIPSCs had a slower peak of decay *τ* (31.7 ± 1.3 ms; Figure 1D). Notably, the decay *τ* of the fast monoexponential mIPSCs was, on average, a factor of 3.6 faster than that of the slow monoexponential mIPSCs. In addition, there was another population of mIPSCs with a slow decay phase, which required fitting with two exponentials. Biexponential mIPSCs yielded components that had a fast (τ_1_ = 7.9 ± 3.3 ms) and a slow (τ_2_ = 116.7 ± 40.2 ms) component (data not shown). The decay τ_1_ was thus a factor of 14.8 faster than the decay τ_2_. These data show that, according to decay kinetics, three types of mIPSCs can be recorded from MDH lamina II_i_ PKCγ^+^ interneurons: fast monoexponential, slow monoexponential, and biexponential mIPSCs. The same decay kinetics were discriminated when sIPSCs were analyzed (Table 3).

A comparison of the other functional properties of the fast monoexponential, slow monoexponential, and biexponential mIPSCs indicated that the three types of mIPSCs were not significantly different in rise time (Figure 1E) or in amplitude (Figure 1F). Finally, the three types of mIPSCs exhibited the same instantaneous frequencies (Table 3).

#### 3.1.2. Co-Occurrence of Fast Monoexponential, Slow Monoexponential, and Biexponential Spontaneous IPSCs in All MDH Lamina II_i_ PKCγ^+^ Interneurons

We then asked whether all PKCγ^+^ interneurons exhibit the three types of IPSCs: fast monoexponential, slow monoexponential, and biexponential IPSCs. Since the same three kinetically defined types of sIPSCs (*n* = 5) or mIPSCs (*n* = 5) could be recorded from MDH lamina II_i_ PKCγ^+^ interneurons (Table 3), we examined the proportions of each of them. On average, 34.5% ± 5.4% (fast monoexponential), 28.9% ± 4.2% (slow monoexponential), and 36.7% ± 3.3% (biexponential) of the total number of spontaneous IPSCs were displayed by MDH lamina II_i_ PKCγ^+^ interneurons (Figure 2). These results thus indicate that all MDH lamina II_i_ PKCγ^+^ interneurons are bombarded by the three kinetically defined types of spontaneous IPSCs in rather equal proportions.

#### 3.1.3. Co-Occurrence of Fast Monoexponential, Slow Monoexponential, and Biexponential IPSCs in MDH Lamina II_i_ PKCγ^+^ and PKCγ^−^ Interneurons

MDH lamina II_i_ PKCγ^+^ and PKCγ^−^ neurons are different according to membrane properties and excitatory synaptic inputs [26,39]. To test whether synaptic inhibitions onto the two cell phenotypes are also different, we compared the sIPSCs or mIPSCs recorded from 10 PKCγ^+^ and 10 PKCγ^−^ interneurons. The total proportions of fast monoexponential, slow monoexponential, and biexponential sIPSCs or mIPSCs were similar in both neuronal populations (Figure 2). Nevertheless, there were differences in the total proportions of IPSCs when compared within PKCγ^−^ interneurons (Figure 2). On the other hand, the PKCγ^+^ (*n* = 5) and PKCγ^−^ interneurons (*n* = 5) displayed the same kinetic properties, except for the instantaneous frequency of mIPSC that was significantly lower in PKCγ^+^ interneurons compared to PKCγ^−^ interneurons (Table 3). Altogether, these results thus suggest that the synaptic inhibitions onto lamina II_i_ PKCγ^+^ and PKCγ^−^ interneurons are similar.

#### 3.1.4. GABA_A_R- and GlyR-Mediated sIPSCs in MDH Lamina II_i_ PKCγ^+^ and PKCγ^−^ Interneurons

We finally examined whether these sIPSCs were mediated by activation of GABA_A_Rs or GlyRs by using the selective GABA_A_R antagonist, bicuculline (10 µM), and GlyR antagonist strychnine (0.5 µM). Since sIPSCs and mIPSCs recorded from MDH lamina II_i_ PKCγ^+^ and PKCγ^−^ interneurons displayed the same kinetic properties, the effects of bath-applied bicuculline and strychnine were assessed on sIPSCs recorded from four lamina II_i_ interneurons (two PKCγ^+^ and 2 PKCγ^−^ interneurons) and result with PKCγ^−^ interneurons were compared to those from PKCγ^+^ interneurons (Figure 3). Bicuculline blocked a large population of sIPSCs (76.3% ± 1.3% for PKCγ^+^ and 75.6% ± 1.3% for PKCγ^−^), and the remaining bicuculline-insensitive sIPSCs were all abolished by the GlyR antagonist, strychnine, indicating that the latter sIPSCs were mediated via activation of GlyRs (Figure 3A). Notably, all lamina II_i_ neurons displayed bicuculline-sensitive as well as bicuculline-insensitive, strychnine-sensitive sIPSCs, implying that they all received both GABA_A_R- and GlyR-mediated inhibition.

The decay phase of individual GlyR-mediated sIPSCs could be fit by a monoexponential function (τ = 8.9 ± 0.9 ms). Therefore, these results suggest that GlyR-mediated sIPSCs represent a population of fast monoexponential sIPSCs, while slow monoexponential sIPSCs (τ > 20 ms) are mediated by GABA_A_R in both PKCγ^+^ and PKCγ^−^ lamina II_i_ interneurons within the rat MDH. This is in agreement with previous findings obtained in MDH [32] and SDH lamina I–II neurons [27,32,33,35,37,52], where GlyR- and GABA_A_R-mediated IPSCs were shown to exhibit rapid and slow decay time courses, respectively.

We conclude that at least the vast majority of quantal events (about 70%) onto lamina II_i_ PKCγ^+^ as well as PKCγ^−^ interneurons within the rat MDH are mediated by GlyR or GABA_A_R only, but not by both. Moreover, all lamina II_i_ interneurons receive GlyR as well as GABA_A_R-only-mediated inputs.

### 3.2. Inhibitory Receptors and Synapses onto PKCγ^+^ Interneurons

To assess the presence of GABA_A_R and/or GlyR onto lamina IIi PKCγ^+^ interneurons, we labeled MDH sections with antibodies against the α2 subunit of GABA_A_R (GABA_A_Rα2), the α1 subunit of GlyR (GlyRα1), and PKCγ.

Triple-immunofluorescence showed that the soma of PKCγ^+^ interneurons (25 PKCγ^+^ interneurons/rat in three rats) received a high density of inhibitory synapses: the number of receptor-IR puncta that were identified on the somata of the PKCγ^+^ cells ranged from 3 to 28 with an average of 11. In most PKCγ^+^ interneurons (49 of 75 cells, 65%; Figure 4A,B), all puncta were labeled for both GABA_A_Rα2 and GlyRα1. Nevertheless, a small number of PKCγ^+^ interneurons had only single-labeled puncta: either GlyRα1-only (5 of 75 cells, 7%) or GABA_A_Rα2-only (2 of 75 cells, 3%) puncta (Figure 4B). Each of the remaining 19 cells displayed double-labeled puncta together with either GlyRα1-only (13 of 75 cells, 17%) or GABA_A_Rα2-only (6 of 75 cells; 8%) puncta (Figure 4B). Pooling data from all 75 cells analyzed, we identified 843 receptor-IR puncta, of which 660 (78.3%) were positive for both GlyRα1 and GABA_A_Rα2, 164 (19.5%) were positive for only GlyRα1, and 19 (2.3%) were positive for only GABA_A_Rα2. Considering each subunit, we identified 824 GlyRα1 puncta, of which 660 (80%) were also GABA_A_Rα2-IR, and 679 GABA_A_Rα2 puncta, of which 660 (97%) were also GlyRα1-IR. Of note, we never observed single-labeled GlyRα1 and GABA_A_Rα2 puncta on the same PKCγ interneuron. This suggests that most PKCγ^+^ interneurons (68 of 75 cells; 91%) receive mixed GABA_A_R–GlyR inhibitory synapses. Some PKCγ^+^ interneurons also receive GABA_A_R-only or GlyR-only synapses, alone or together with mixed GABA_A_R–GlyR synapses. Our analysis of PKCγ dendrites yielded quantitatively similar results. We identified a mean of 106.7 (range 102–115) GABA_A_Rα2 puncta on PKCγ dendrites, of which 96.9% (92.8–100%) were also GlyRα1-IR, and 104 (100–111) GlyRα1 puncta, of which 76% (57.3–96.1%) were also GABA_A_Rα2-IR.

We also looked for appositions of GABAergic and glycinergic boutons onto lamina IIi PKCγ^+^ interneurons. Analysis of MDH transverse sections that had been reacted with antibodies against GlyT2, GADs, gephyrin, and PKCγ revealed that both the somata and dendrites of PKCγ^+^ interneurons had numerous gephyrin puncta, which represent the sites of inhibitory synapses (Figure 4C) [47]. We analyzed between 73 and 86 (mean 77.7) gephyrin puncta that were on cell bodies or dendrites of PKCγ^+^ cells in sections from the three rats. We found that 42.2% of these (range 37.8–47.9%) were adjacent to boutons that were positive for both GAD and GlyT2, while 49.5% (range 46.5–54.1%) were in contact with GAD-positive boutons that lacked GlyT2, and 6.3% (range 4.1–9.3%) were contacted by a bouton that was GAD-negative and GlyT2-positive. In the remaining 2.1% (range 0–3.5%) of cases, we did not observe structures with either GAD-IR or GlyT2-IR. This suggests that most inhibitory boutons that synapse on the PKCγ^+^ cells release GABA, with around half also being glycinergic.

## 4. Discussion

We performed electrophysiology and immunochemistry to characterize synaptic inhibition onto lamina II_i_ PKCγ^+^ interneurons in the adult rat MDH. According to decay time courses, three distinct populations of sIPSCs or mIPSCs can be recorded from these lamina II_i_ PKCγ^+^ interneurons: fast monoexponential, slow monoexponential, and biexponential sIPSCs or mIPSCs. All three types of sIPSCs or mIPSCs are present in all recorded PKCγ^+^ with rather similar frequencies. Moreover, the same quantal events are recorded from neighboring lamina II_i_ PKCγ^−^ interneurons, suggesting that lamina II_i_ PKCγ^+^ and PKCγ^−^ interneurons display similar types of inhibitory inputs. The sensitivity of fast and slow quantal events to GABA_A_R and GlyR antagonists suggests that monoexponential sIPSCs or mIPSCs are mediated by GlyR and GABA_A_R, respectively. Altogether, our electrophysiological results suggest that inhibitory inputs onto lamina II_i_ PKCγ^+^ or PKCγ^−^ interneurons in the adult rat MDH are predominantly (at least 70%) mediated by either GlyR or GABA_A_R but not both. On the other hand, immunohistochemistry reveals that the vast majority (~80%) of inhibitory synapses onto PKCγ^+^ interneurons contain both GABA_A_Rs and GlyRs. We also show that the great majority (~90%) of inhibitory boutons presynaptic to the PKCγ^+^ cells possess GAD and can therefore presumably release GABA, while just under half of these also express GlyT2 and are therefore likely to co-release glycine.

### 4.1. GABA_A_R-Only- or GlyR-Only-Mediated IPSCs onto MDH lamina II_i_ PKCγ^+^ and PKCγ^−^ Interneurons

Two populations of monoexponential sIPSCs or mIPSCs (fast and slow monoexponential sIPSCs or mIPSCs ) were recorded from lamina II_i_ PKCγ^+^ interneurons in the adult rat MDH. Fast monoexponential sIPSCs or mIPSCs are mediated by GlyRs. Their decay *τ* is similar to that of bicuculline-resistant, strychnine-sensitive IPSCs (present results) and within the range of those of previously recorded GlyR-mediated IPSCs from SDH/MDH lamina I–II neurons [27,31,32,34,35,36]. GABA_A_Rs mediated the slow monoexponential sIPSCs or mIPSCs as long-duration IPSCs were all suppressed by bath-applied bicuculline (present results). Moreover, their decay *τ* is within the range of those of previously recorded GABA_A_R-mediated IPSCs from SDH/MDH lamina I–II neurons [27,31,32,34,35,36].

Two types of biexponential-evoked IPSCs and mIPSCs were previously recorded from lamina II neurons: pure GABA_A_R-mediated within the MDH [32] and mixed GABA_A_R/GlyR IPSCs (that is, combining two components, a fast decaying, GlyR-mediated and a slowly decaying, GABA_A_R-mediated) in the SDH [35,36]. The biexponential sIPSCs or mIPSCs that we recorded from MDH lamina II_i_ PKCγ^+^ interneurons at post-natal day (P) 21–35 are unlikely to be mixed GABA_A_R/GlyR IPSCs since these very IPSCs, in SDH lamina II neurons, are only present during early development and become virtually undetectable by approximately P23 [35,37]. They are rather GABA_A_R-mediated. First, similar biexponential GABA_A_R-mediated mIPSCs were already recorded in this very area in juvenile to adult rats [32]. Second, compared with slow monoexponential, GABA_A_R-mediated mIPSCs, biexponential mIPSCs exhibit larger amplitudes and longer decay *τ* of the slow component (here and in the work of [32]). Finally, bath application of bicuculline strongly reduced sIPSCs frequency (more than 2/3 reduction), consistent with the observation that GABA_A_R-mediated mIPSCs occur at a higher frequency than GlyR-mediated mIPSCs in MDH lamina II neurons [32]. The proportion of GlyR-only sIPSCs recorded from PKCγ^+^ interneurons in the presence of bicuculline (~24%) is in the same range as that of kinetically isolated fast monoexponential IPSCs (~34%). Indeed, if our biexponential sIPSCs were mixed GABA_A_R/GlyR IPSCs, the proportion of GlyR-only sIPSCs in the presence of bicuculline should have been double; that is, the sum of the proportions of kinetically isolated fast monoexponential IPSCs and biexponential IPSCs. Therefore, our electrophysiological data strongly suggest that all quantal inhibition onto lamina II_i_ PKCγ^+^ interneurons in adult rat MDH is mediated by either GlyR or GABA_A_R but not by both types of receptors concurrently.

This conclusion holds true for lamina II_i_ PKCγ^−^ neurons as the same quantal events were recorded from PKCγ^+^ and PKCγ^−^ interneurons. Therefore, whereas these two classes of lamina II_i_ interneurons exhibit differences in their membrane properties and excitatory inputs [26,39], they receive similar inhibitory inputs, at least in the adult rat MDH.

Altogether, our results are consistent with the conclusion that lamina II neurons in the adult rat MDH [32] as well as SDH [27,31,32,34,35,36] exhibit quantal events mediated by GlyR and by GABA_A_R, but not by both types of receptors concurrently. However, whereas GABA_A_R-only IPSCs in the SDH are exclusively monoexponentially decaying, those in the MDH can be mono- as well as biexponentially decaying (present results; [32]). It is interesting to note that the decay *τ* of the fast component of biexponential mIPSCs is within the same range as that of GABA_A_R-mediated mIPSCs recorded from pyramidal neurons in the rat somatosensory cortex [53]. Such fast GABA_A_R-mediated mIPSCs might be related to the expression of the GABA_A_Rα1 subunit [54]. Thus, with respect to GABA_A_R-mediated inhibition, MDH would be intermediate between the SDH and higher brain areas.

### 4.2. PKCγ^+^ Interneurons Co-Express GABA_A_R and GlyR at the Same Inhibitory Synapses

The present results establish that the subunits belonging to GABA_A_R and GlyR are colocalized at the great majority (~80%) of inhibitory synapses on PKCγ^+^ cells and that nearly all (91%) of these cells have synapses at which the receptors are co-expressed. Nevertheless, some PKCγ^+^ interneurons (9%) also displayed synapses with only GABA_A_Rs or GlyRs, which may reflect the existence of different subpopulations of PKCγ^+^ interneurons [26,39].

However, we have to consider certain technical issues. We have previously demonstrated with confocal microscopy that virtually all GlyRα1-IR puncta in the rat spinal dorsal horn are also immunoreactive for gephyrin [30]. We have also shown that gephyrin puncta seen at the light microscope level are invariably associated with inhibitory axonal boutons identified by expression of the vesicular GABA transporter (which also transports glycine) [47] and that they correspond to the postsynaptic specialization of inhibitory synapses when viewed with electron microscopy [55]. It is, therefore, very likely that all of the GlyRα1 puncta that we observed in the membranes of PKCγ cells represent inhibitory synapses, and as we show here, most of these (80%) were also GABA_A_Rα2-IR. The GABA_A_Rα2 is also expressed by primary afferent axons, and so some of the immunostaining for this receptor might have been associated with primary afferent terminals. However, this is highly unlikely to account for any of the GABA_A_Rα2 puncta that we identified on PKCγ neurons because the vast majority of these puncta (98%) also contained the GlyRα1 subunit, which is not expressed by primary afferents [56,57,58].

Negative findings with immunohistochemistry (i.e., lack of immunoreactivity for one or other receptor subunit at individual puncta) could result from low levels of the receptor (rather than its absence) at a synapse. Indeed, “single-labeled” puncta were often quite weakly stained for either GABA_A_R or GlyR, so they may have had low levels of the other receptor that were below the detection threshold. Another possibility is that a synapse may only express other subunits for the receptor. We assessed the expression of GABA_A_Rα2 and GlyRα1 on lamina II_i_ PKCγ^+^ interneurons because these subunits are known to be abundant in SDH [59,60,61]. For the few cells that completely lacked GlyRα1, it is unlikely that they have functional glycine receptors (although they could presumably express GlyRα3) [62]. However, for cells that lack the GABA_A_Rα2 subunit, it is quite possible that they express another GABA_A_ α subunit. It is thought that individual GABA_A_ receptors may contain two different α subunits [44], and it is also possible that individual neurons may possess receptors that contain different subunits (e.g., α2 or α3). Thus, we might have underestimated the proportion of cells and synapses that express both GABA_A_R and GlyR.

Previous immunohistochemical studies showed that 30–50% of superficial SDH neurons are GABAergic, and about half of these are also enriched with glycine [63,64,65]. This is consistent with our finding that most inhibitory boutons presynaptic to PKCγ^+^ interneurons stained for GAD, while around 40% of them co-expressed GlyT2.

Detection of synaptic receptors required pepsin treatment, which presumably exposes epitopes that are normally masked by fixation [40,66]. This treatment also destroys many other antigens [40], and to avoid this problem, we used tyramide signal amplification to reveal PKCγ. However, it is difficult to combine antigen unmasking and receptor immunostaining with the detection of more than other antigens, and so we were unable to directly compare the expression of presynaptic markers (GADs, GlyT2) with that of postsynaptic receptors on the PKCγ^+^ cells.

### 4.3. Can Anatomical Data Account for the Functional Specialization of Inhibitory Synapses onto MDH Lamina II_i_ PKCγ^+^ Interneurons?

We can make certain predictions concerning the arrangement of neurotransmitters and receptors based on our anatomical finding that ~80% of inhibitory synapses had both GABA_A_Rα2 and GlyRα1 subunits and that >90% of presynaptic boutons were GAD^+^, whereas only around half were GlyT2^+^. Thus, many inhibitory synapses onto PKCγ^+^ interneurons may have both types of receptors but only involve GABA release. The few synapses that only express GABA_A_Rs should also operate as purely GABAergic. Similarly, inhibitory synapses that have only GlyRs should function as purely glycinergic.

However, GlyR-only inhibitory synapses, as revealed by immunohistochemistry, are unlikely to account for the 34% of GlyR-only-mediated sIPSCs or mIPSCs, even though such synapses would display high release probabilities. Moreover, our anatomical data suggest that, in numerous inhibitory synapses onto PKCγ^+^ interneurons, GABA and glycine are co-released, and GlyR and GABA_A_R are both present on the postsynaptic membrane. One would therefore expect mixed GABA_A_R/GlyR-mediated mIPSCs to be recorded. However, our electrophysiological data show that, in normal conditions, lamina II_i_ PKCγ^+^ interneurons in adult rat MDH are exclusively bombarded by GABA_A_R-only- and GlyR-only-mediated IPSCs, similar to lamina II neurons in young rat SDH [27,31,32,34,35,36]. Recently, while recording ex vivo from lamina II_i_ PKCγ^+^ interneurons in 6–8-week-old mice, Wang et al. [26] observed not only pure Aβ fiber-evoked GlyR-mediated IPSCs in most neurons (7 out of 12) but also mixed Aβ fiber-evoked GABA_A_R-GlyR-mediated IPSCs in the other five. However, in this report, dorsal roots were electrically stimulated, and therefore, many Aβ fibers were activated simultaneously: the few mixed GABA_A_R-GlyR-mediated IPSCs might thus result from summated Aβ fiber-evoked pure GABA_A_R- and GlyR-mediated IPSCs. Altogether, this suggests that some synapses can function as purely GABAergic or glycinergic even though presynaptic buttons co-release GABA/glycine and postsynaptic membranes contain both GABA_A_ and glycine receptors. One possibility is that GABA_A_R and GlyR are differentially located within inhibitory synapses. Indeed, bath application of a benzodiazepine [27] or inflammatory conditions [37] leads to the unmasking of a long-duration GABA_A_R-mediated component and the appearance of mixed GABA_A_R/GlyR mIPSCs in adult SDH lamina II neurons. This might stem from a different localization of GABA_A_R and GlyR in inhibitory synapses onto SDH neurons, with GABA_A_Rs being extrasynaptically distributed [27]. However, our anatomical data in adult rat MDH are not consistent with this hypothesis, as the two types of receptors were co-extensive at presumed inhibitory synapses. A more likely explanation is that although there is co-expression of the GABA_A_Rα2 and GlyRα1 subunits at these synapses, one or another type of receptor is not functional, perhaps related to the differential expression of other subunits.

The present finding of GABA_A_R-only- and GlyR-only-mediated sIPSCs or mIPSCs recorded from PKCγ^+^ interneurons suggests separate roles for these two inhibitory systems. There are two clinical forms of mechanical allodynia: dynamic and static. Whereas both types are impacted by modulating PKCγ activity [5,6,14,16], glycinergic disinhibition produces specifically dynamic mechanical allodynia [5] and GABA_A_ergic disinhibition, a static one [6,22]. How can mechanical hypersensitivity depend on the type of disinhibition? LTMRs transmitting the dynamic (Aβ-LTMRs) and static (Aδ-LTMRs) mechanical inputs to PKCγ^+^ interneurons might be gated by specifically glycinergic and GABA_A_ergic feedforward inhibitory microcircuits, respectively (see Figure 1 in the work of [9]). There is already evidence for feedforward glycinergic circuits preventing the activation of PKCγ^+^ interneurons and superficial nociceptive circuits by Aβ inputs, which is reduced following nerve injury [20,26]. It is, therefore, possible that differential modulation of glycinergic and GABA_A_ergic feedforward inhibitions onto PKCγ^+^ interneurons might be key determinants of the manifestation of dynamic and static mechanical allodynias, respectively.

## 5. Conclusions

Although our anatomical data suggest that GABA_A_R and GlyR are colocalized at most inhibitory synapses on these cells and that GABA and glycine are often co-released, PKCγ^+^ interneurons exclusively displayed GABA_A_R-only- and GlyR-only-mediated spontaneous IPSCs. Behavioral evidence suggests that this specialization of inhibitory inputs onto PKCγ^+^ interneurons is functionally relevant.

Data represent F-values for one or two-way ANOVA with corresponding degrees of freedom (DFn, DFd) and P-values. Factors for ANOVA were designed as follows: Cell type, for comparisons between PKCγ+ and PKCγ–; IPSC type for comparisons between fast monoexponential, slow monoexponential, and biexponential mIPSC; and Interaction, for effect between two factors on the dependent variable.

## Figures and Tables

**Figure 1 cells-11-01356-f001:**
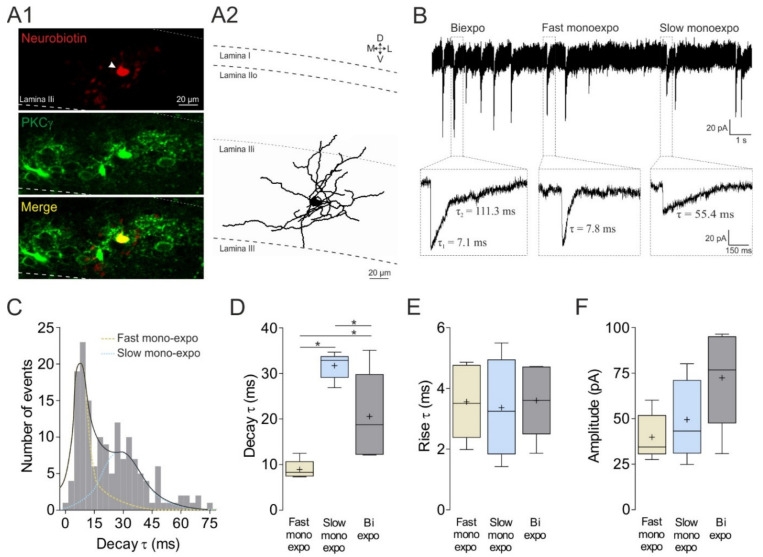
Functional properties of mIPSCs recorded from lamina II_i_ PKCγ^+^ interneurons. (**A**) PKCγ^+^ neurobiotin-filled interneuron. Confocal images showing the neurobiotin labeling of a recorded neuron (red, arrowhead) that colocalized with the PKCγ immunostaining (green; (**A1**)). Immuno-labeling was performed in parasagittal slices (350 µm thick). Representative neuronal reconstruction showing the neuritic arborization of recorded PKCγ^+^ interneuron (**A2**). M, medial; L lateral; D, dorsal; V, ventral. Dashed lines represent laminae limits. (**B**) (Top) Whole-cell patch-clamp recordings (voltage-clamp mode; holding potential: −65 mV) of miniature inhibitory postsynaptic currents (mIPSCs) from a PKCγ^+^ interneuron (in the presence of CNQX (10 μM) and APV (40 μM)). (Bottom) Magnified events of the three categories of mIPSCs (on the basis of their decay kinetics): left, slowly biexponentially decaying mIPSC (Biexpo); middle, fast monoexponentially decaying mIPSC (Fast monoexpo); right, slowly monoexponentially decaying mIPSC (Slow monoexpo). Each computed tau (τ) value is indicated close to the corresponding component. (**C**) Close inspection of the kinetics of the monoexponentially decaying mIPSC from PKCγ^+^ interneurons revealed a heterogeneous population of decay times, which were best fitted by the sum of two Gaussians. (**D**–**F**) Boxplots of the decay τ (**D**), the rise τ (**E**), and the amplitude (**F**) of the fast monoexponentially decaying, the slowly monoexponentially decaying, and the slowly biexponentially decaying mIPSCs; the *τ* values of the latter were obtained by forcing monoexponential fits. Data are presented as box-and-whisker plots depicting median, interquartile interval, minimum, and maximum. The “+” represents the medians, and the boxes present the quartiles. Data were averaged from 5 lamina II_i_ PKCγ^+^ interneurons. The comparison between groups was made using one-way ANOVA followed by Tukey’s *post-hoc* test. * *p* < 0.05.

**Figure 2 cells-11-01356-f002:**
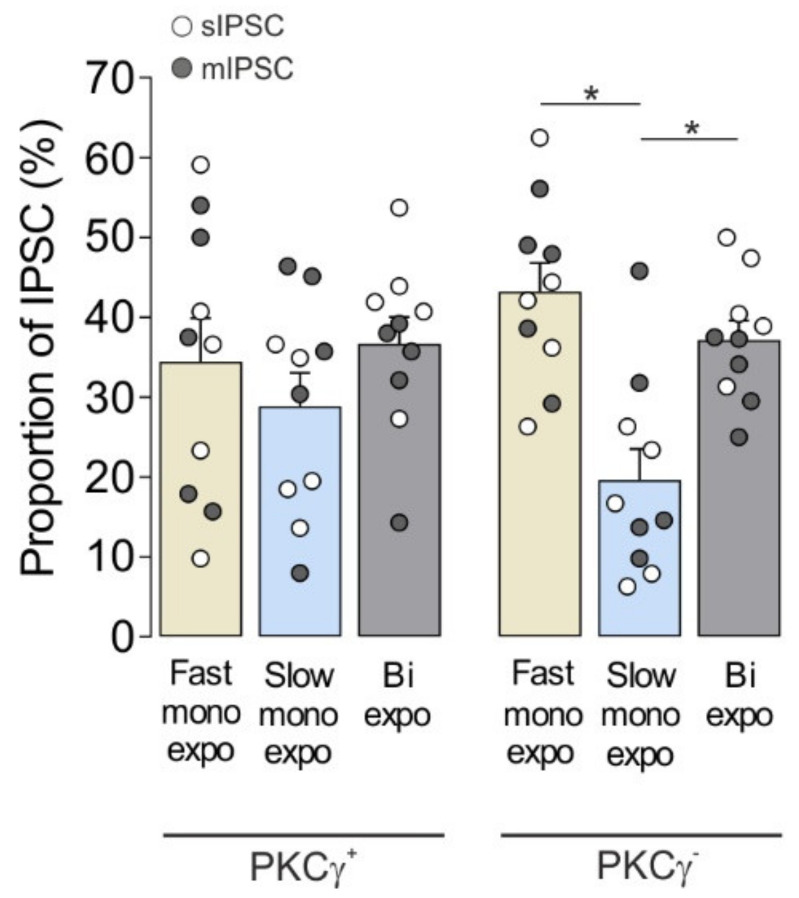
Proportions of kinetically defined sIPSC and mIPSCs recorded from lamina IIi PKCγ^+^ and PKCγ^−^ interneurons. Bar histogram showing the mean ± SEM of proportions (%) of the different kinetically defined (fast monoexponential, slow monoexponential, and biexponential) spontaneous (sIPSC) or miniature inhibitory postsynaptic currents (mIPSC) recorded from lamina II_i_ PKCγ^+^ (*n* = 10) or PKCγ^−^ (*n* = 10) interneurons. The comparison between groups was made using two-way ANOVA followed by Tukey’s *post-hoc* test. * *p* < 0.05.

**Figure 3 cells-11-01356-f003:**
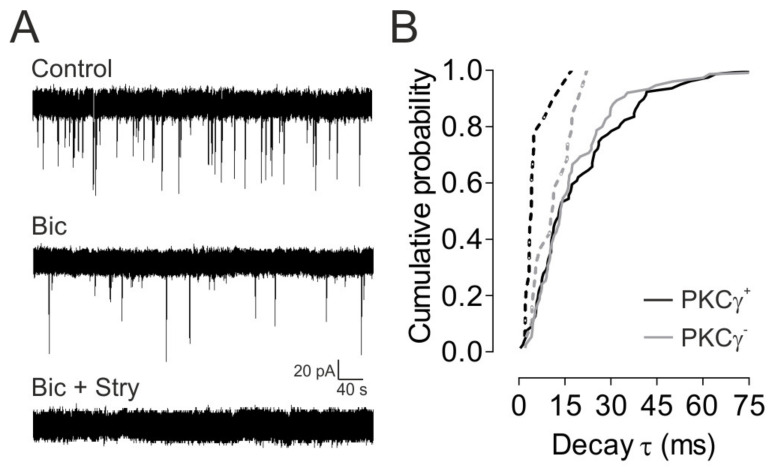
GlyR- and GABA_A_R-mediated sIPSCs in lamina II_i_ PKCγ^+^ and PKCγ^−^ interneurons. (**A**) Whole-cell patch-clamp recordings (voltage-clamp mode; holding potential: −65 mV) of spontaneous inhibitory postsynaptic currents (sIPSCs) from a lamina II_i_ PKCγ^+^ interneuron in control condition (in the presence of CNQX (10 μM) and APV (40 μM)) and after application of bicuculline (Bic; GABA_A_R antagonist, 10 μM) and strychnine (Stry; GlyR antagonist, 0.5 µM). Note that sIPSC were strongly reduced (number) after bicuculline and completely abolished after bicuculline + strychnine. (**B**) Cumulative probability plots of decay *τ* of sIPSCs recorded from 4 lamina II_i_ neurons, 2 PKCγ^+^ (black), and 2 PKCγ^−^ (gray) neurons. The plots were constructed by forcing monoexponential fits to all individual sIPSCs recorded under control conditions (solid lines) and then in the presence of bicuculline (dashed lines). Note that bicuculline abolished all events with decay *τ* higher than 20 ms.

**Figure 4 cells-11-01356-f004:**
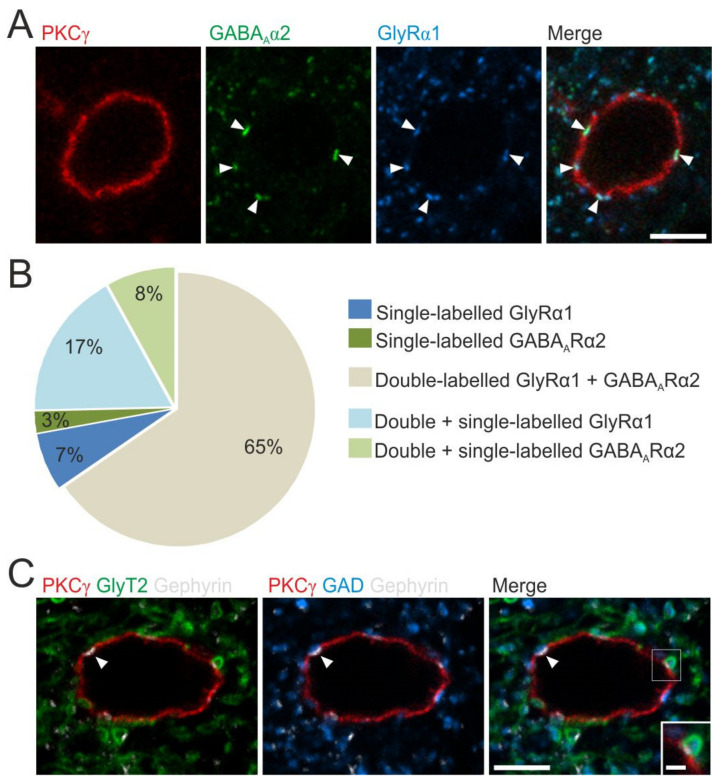
Inhibitory receptors and synapses on lamina II_i_ PKCγ^+^ interneurons. (**A**) Confocal images showing a PKCγ^+^ interneuron (red) with GABA_A_Rα2 (green) and GlyRα1 (blue) labelings. The merged image shows that these inhibitory receptor subunits were colocalized on the cell body (arrowheads). Scale bar = 5 µm. (**B**) Proportions of PKCγ^+^ interneurons that display GABA_A_ergic and/or glycinergic receptors subunits (*n* = 75). (**C**) Confocal image of a single optical section through a part of lamina II_i_, scanned to reveal PKCγ (red), gephyrin (white), GlyT2 (green), and GAD (blue). Gephyrin labeling indicates the localization of inhibitory synapses on the PKCγ cell body, and GlyT2 and GAD labelings show the terminal synaptic boutons. The arrowhead indicates a synapse from a GAD-positive GlyT2-negative bouton, while the inset (corresponding to the box magnification, scale bar = 1 µm) shows a synapse from a bouton that is positive for both GAD and GlyT2 (scale bar = 5 µm).

**Table 1 cells-11-01356-t001:** Summary of statistical analysis.

Figures	Analysis (*Post-Hoc* Test)	Factor Analyzed	F-Ratios	*p*-Values
Figure 1D	One-way ANOVA (Tukey)	IPSC type	IPSC type F(2,12) = 18.4	**<0.001**
Figure 1E	One-way ANOVA (Tukey)	IPSC type	IPSC type F(2,12) < 0.1	1.0
Figure 1F	One-way ANOVA (Tukey)	IPSC type	IPSC type F(2,12) = 3.1	0.08
Figure 2	Two-way ANOVA (Tukey)	Cell type × IPSC type	Cell type F(1,54) < 0.1	1.0
			IPSC type F(2,54) = 8.2	**<0.001**
			Interaction F(2,54) = 2.6	0.08

**Table 2 cells-11-01356-t002:** Summary of statistical analysis.

Variables	Analysis (*Post-Hoc* Test)	Factor Analyzed	F-Ratios	*p*-Values
Decay τ of mIPSC (ms)	Two-way ANOVA (Tukey)	Cell type × IPSC type	Cell type F(1,24) = 0.4	0.5
			IPSC type F(2,24) = 42.4	**<0.001**
			Interaction F(2,24) = 1.3	0.3
Decay τ of sIPSC (ms)	Two-way ANOVA (Tukey)	Cell type × IPSC type	Cell type F(1,24) = 0.3	0.6
			IPSC type F(2,24) = 46.8	**<0.001**
			Interaction F(2,24) = 0.1	0.5
Rise τ of mIPSC (ms)	Two-way ANOVA (Tukey)	Cell type × IPSC type	Cell type F(1,24) = 2.5	0.1
			IPSC type F(2,24) < 0.1	1.0
			Interaction F(2,24) = 0.2	0.8
Instantaneous frequency of mIPSC (Hz)	Two-way ANOVA (Tukey)	Cell type × IPSC type	Cell type F(1,24) = 6.6	**<0.05**
			IPSC type F(2,24) = 0.2	0.8
			Interaction F(2,24) = 0.2	0.8
Instantaneous frequency of sIPSC (Hz)	Two-way ANOVA (Tukey)	Cell type × IPSC type	Cell type F(1,24) < 0.1	0.8
			IPSC type F(2,24) = 0.5	0.6
			Interaction F(2,24) = 0.6	0.6
Amplitude of mIPSC (pA)	Two-way ANOVA (Tukey)	Cell type × IPSC type	Cell type F(1,24) < 0.1	0.9
			IPSC type F(2,24) = 5.1	**<0.05**
			Interaction F(2,24) = 0.3	0.8
Amplitude of sIPSC (pA)	Two-way ANOVA (Tukey)	Cell type × IPSC type	Cell type F(1,24) = 0.4	0.5
			IPSC type F(2,24) = 0.8	0.4
			Interaction F(2,24) < 0.1	0.9

**Table 3 cells-11-01356-t003:** Functional properties of mIPSC or sIPSCs recorded from lamina II_i_ PKCγ^+^ and PKCγ^−^ interneurons.

	PKCγ^+^		PKCγ^−^	
	mIPSC (5)	sIPSC (5)	mIPSC (5)	sIPSC (5)
**Decay τ (ms)**				
Fast monoexpo	8.9 ± 0.9 ^a,b^	10.0 ± 1.0	11.7 ± 0.6	11.8 ± 1.4
Slow monoexpo	31.7 ± 1.3 ^c^	31.9 ± 2.0	30.1 ± 0.7	30.4 ± 2.4
Biexpo	20.6 ± 4.3	19.6 ± 2.7	16.0 ± 2.9	16.6 ± 2.6
**Rise τ (ms)**				
Fast monoexpo	3.6 ± 0.6	~	2.4 ± 0.6	~
Slow monoexpo	3.4 ± 0.7	~	3.0 ± 0.6	~
Biexpo	3.6 ± 0.5	~	2.6 ± 0.9	~
**Instantaneous frequency (Hz)**				
Fast monoexpo	0.2 ± 0.1	0.5 ± 0.2	1.1 ± 0.5	0.3 ± 0.1
Slow monoexpo	0.3 ± 0.1	0.2 ± 0.1	0.7 ± 0.5	0.3 ± 0.1
Biexpo	0.3 ± 0.1	0.3 ± 0.1	1.0 ± 0.4	0.6 ± 0.1
**Amplitude (pA)**				
Fast monoexpo	39.9 ± 5.7	37.1 ± 13.2	48.5 ± 7.2	33.1 ± 6.0
Slow monoexpo	49.5 ± 9.8	37.5 ± 8.9	46.9 ± 5.9	35.4 ± 8.1
Biexpo	72.4 ± 11.9	51.7 ± 12.2	69.5 ± 11.7	42.7 ± 11.7

Data show mean ± SEM of decay τ (ms), rise τ (ms), instantaneous frequency (Hz), and amplitude (pA) of fast monoexponential, slow monoexponential, and biexponential miniature (mIPSCs) and spontaneous inhibitory postsynaptic currents (sIPSC) from (n) lamina II_i_ PKCγ^+^ or PKCγ^−^ interneurons. Tau (τ) values were obtained by forcing monoexponential fits. The comparison between groups was made using two-way ANOVA followed by Tukey’s *post-hoc* test. ^a^
*p* < 0.001 Fast monoexp compared to Slow monoexpo PKCγ^+^, ^b^
*p* < 0.05 Fast monoexp compared to Biexpo PKCγ^+^, and ^c^
*p* < 0.05 Slow monoexpo compared to Biexpo PKCγ^+^. ~ Not analyzed.

## Data Availability

All data supporting the conclusions of this manuscript are provided in the text and figures.

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
