# Peer review of "GABA_A_ and Glycine Receptor-Mediated Inhibitory Synaptic Transmission onto Adult Rat Lamina II_i_ PKCγ-Interneurons: Pharmacological but Not Anatomical Specialization"

_cells, 2022, doi:10.3390/cells11081356_

Round 1

Reviewer 1 Report

An interesting work analysing the contribution of GABA and glycine to PKCγ+ interneuron inhibition connecting touch and pain circuits.

Author Response

We thank reviewer for their positive thoughts on our manuscript

Reviewer 2 Report

The authors Khoueiry et al. described PKCgamma(+) interneurons on the lamina IIi of medullary spinal cords received inhibitory postsynaptic currents were pharmacological-dependent, but not anatomical-dependents fashions. Generally, the experimental designs in this manuscript were well-designed and in good preparation and writing. There were some commands:

  1. Anatomically, the peripheral terminals of LTMRs should be in the gracilis or cuneate nucleus of the medullary cord. I suggested the authors should address the specific nucleus area in the immunohistochemical approaches.
  2. According to Fig. 3, what is the IPSCs changes of strychnine followed by bicuculline? And what is the IPSCs changes only by strychnine administration?
  3. Line 116, “7.0 MgCl2” should be “7 MgCl2”.
  4. Line 130, “X10”, line 132, “X63”. Should be “10X” and “63X” are better than originally?
  5. Line 156, It should be added country in the product/reagent information.
  6. Line 251, “63x”, should be “63X”.
  7. Line 253, “Twenty-five” should be “25”?

Author Response

  1. Anatomically, the peripheral terminals of LTMRs should be in the gracilis or cuneate nucleus of the medullary cord. I suggested the authors should address the specific nucleus area in the immunohistochemical approaches.

We do not understand this comment: central (not peripheral!) terminals of large myelinated afferents that enter the spinal cord through dorsal roots do indeed send branches to the gracile and cuneate nucleus. However, these are not directly relevant to this study, which involved investigation of inhibitory inputs to PKCγ-expressing interneurons in the trigeminal nucleus caudalis (medullary dorsal horn). There may have been some confusion because this region was referred to as "MDH" in the electrophysiological part of the Methods section and "spinal trigeminal nucleus" in the anatomical part. To clarify, we have replaced "spinal trigeminal nucleus" with "MDH" throughout the text.

  1. According to Fig. 3, what is the IPSCs changes of strychnine followed by bicuculline? And what is the IPSCs changes only by strychnine administration?

Our aim was to assess the respective contribution of GABA and glycine to synaptic inhibition onto adult MDH lamina IIi PKCγ+ interneurons.

To address this issue, we examined the changes in spontaneous and mini IPSCs (sIPSC and mIPSCs) induced by bath-applied bicuculline followed by strychnine. From these results, we could conclude that all inhibitory quantal events onto adult MDH lamina IIi PKCγ+ interneurons were GlyR-only or GABAAR-only.

We could also have examined sIPSC and mIPSCs changes induced by bath-applied bicuculline followed by strychnine, as suggested by the reviewer. We would have reached the same conclusion!

  1. Line 116, “7.0 MgCl2” should be “7 MgCl2”.
    We have made this change.

  2. Line 130, “X10”, line 132, “X63”. Should be “10X” and “63X” are better than originally?
    We have changed these to the format used for the anatomical part, to be consistent.

  3. Line 156, It should be added country in the product/reagent information.
    We have added the country.

  4. Line 251, “63x”, should be “63X”.
    No, this is correct as it is. "×" is the mathematical sign for "multiplied by", not the alphabetical letter "x".

  5. Line 253, “Twenty-five” should be “25”?
    No, this is correct as it is. Sentences should not start with a numeral "25", this should be spelled out.

Reviewer 3 Report

Comments to the Author

The report by El Khoueiry et al. is an interesting study focusing on the role and mechanism of the

GABAA and Glycine Receptor-Mediated Inhibitory Synaptic. The manuscript is very well and detailed written. Results and methods are clearly formulated. These data should be shared with the community.

Please find some minor concerns below:

  1. “adult male Sprague-Dawley rats (21-35 days old, 50-100 g, n = 13; 42-49 days old,

280-290 g, n = 3) have been involved into study. On what basis this age of animals was chosen, and not, for example, 3 months ? The second group n=3 is minimal.

  1. Table 1. Functional properties of mIPSC or sIPSCs. Why the Rise (ms) of sIPSCs for the PKCγ+/- interneurons were not analysed ?
  2. The overview table of PKCγ+/-, GlyR- and GABA receptor expression, brain region/structures, function, inhibition and stimulation in animal model and in human is missing for better clarification of the contents.
  3. The discussion lacks translation for the disease unit in the clinic.

Author Response

  1. “adult male Sprague-Dawley rats (21-35 days old, 50-100 g, n = 13; 42-49 days old, 280-290 g, n = 3) have been involved into study. On what basis this age of animals was chosen, and not, for example, 3 months ? The second group n=3 is minimal.

For the electrophysiological part: on the one hand, recordings from ex vivo neurons are better in slices from young animals. On the other hand, we wanted to record from adult PKCγ+ interneurons. We recently assessed the postnatal anatomical and functional development of PKCγ+ interneurons in the rat MDH (Mermet-Joret et al., 2021, doi: 10.1097/j.pain.0000000000002459). We demonstrated that PKCγ+ interneurons have reached their adult phenotype at postnatal day 21. Therefore, we recorded from MDH lamina IIi PKCγ+ interneurons in ex vivo slices obtained from 21-35 days old male Sprague-Dawley rats

For the anatomical part: 6-7 weeks old is fairly standard for anatomical studies, as neuronal circuitry is likely to be fully mature at this stage. The use of 3 animals for the anatomical studies is consistent with the principles of "Reduction, Refinement, Replacement" for animal studies. Large numbers of cells can be obtained from each animal, and there is minimal variability between results obtained from the different animals.

  1. Table 1. Functional properties of mIPSC or sIPSCs. Why the Rise (ms) of sIPSCs for the PKCγ+/- interneurons were not analyzed ?

Only the rise times (ms) of mIPSCs recorded from PKCγ+ and PKCγ interneurons were analyzed.

Spontaneous and mini IPSCs are the same, the latter only being recorded in the presence of TTX and thus action potential-independent. The rise times of sIPSC and mIPSCs should thus be the same.

Moreover, GABAAR only- or GlyR only-mediated mIPSCs exhibit the same rise-times, in the present results (see Table 1) as well as in others (see Keller et al., 2001; doi: 10.1523/JNEUROSCI.21-20-07871.2001). Therefore, sIPSC rise times cannot be used to dissociate between the different pharmacological types of IPSCs.

  1. The overview table of PKCγ+/-, GlyR- and GABA receptor expression, brain region/structures, function, inhibition and stimulation in animal model and in human is missing for better clarification of the contents.
    It is not clear what is wanted here!

  2. The discussion lacks translation for the disease unit in the clinic.

At the end of the Discussion (lines 540-554), we addressed the issue of the specificity of mechanical allodynia – dynamic and static – according to the type of desinhinbition – glycinergic and GABAAergic, respectively, ... at least in animal models of pain. We finish by suggesting that: ‘... differential modulation of glycinergic and GABAAergic feed-forward inhibitions onto PKCγ+ interneurons might be key determinants of the manifestation of dynamic and static mechanical allodynias, respectively.